# Simultaneous Stimulation of Peripheral Blood Mononuclear Cells with CpG ODN2006 and α-IgM Antibodies Leads to Strong Immune Responses in Monocytes Independent of B Cell Activation

**DOI:** 10.3390/cells13221822

**Published:** 2024-11-05

**Authors:** Leonie Fleige, Silvia Capellino

**Affiliations:** Research Group of Neuroimmunology, Department of Immunology, IfADo-Leibniz Research Centre for Working Environment and Human Factors, Ardeystraße 67, 44139 Dortmund, Germany; fleige@ifado.de

**Keywords:** α-IgM, CpG ODN2006, cytokine secretion, monocytes, peripheral blood mononuclear cells

## Abstract

CpG ODN2006 is widely used both in vitro and in vivo to achieve B cell activation and has been previously applied in clinical trials as an adjuvant and anti-cancer agent. Recent studies have demonstrated the benefit of combining CpG ODN2006 with α-IgM antibodies to obtain optimal B cell activation in vitro. In this study, we expanded the knowledge of how both agents affect other types of peripheral blood mononuclear cells (PBMCs), thereby highlighting beneficial and potentially unfavorable properties of the combination of CpG ODN2006 and α-IgM when applied beyond isolated B cells. We elucidated the effects of both compounds on mixed PBMCs, as well as on B cell- and monocyte-depleted PBMCs, allowing us to distinguish between direct effects and indirect influences mediated by other interacting immune cells. Flow cytometry was used to measure the expression of surface markers and intracellular cytokines, while ELISA and multiplex assays were performed to determine cytokine secretion. Our results revealed that stimulation of mixed PBMCs with CpG ODN2006 and α-IgM strongly increased cytokine secretion, primarily originating from α-IgM-stimulated monocytes. Monocyte activation was confirmed by increased CD86 and HLA-DR expression and occurred independently of B cells. The high level of monocyte-derived cytokines after α-IgM exposure did not affect B cell activation. However, it represents a rather unfavorable property for clinical applications. In conclusion, α-IgM is a potent inducer of cytokine production in monocytes. Based on our findings we hypothesize that significant side effects on monocytes can occur when using α-IgM to enhance CpG ODN2006’s efficacy on B cells, particularly in clinical settings.

## 1. Introduction

During recent decades, CpG oligodeoxynucleotides (CpG ODNs) have become widely used tools for activating different types of innate and adaptive immune cells via binding to the endosomal Toll-like receptor 9 (TLR9). The single-stranded ODNs mimic bacterial DNA due to their unmethylated CpG motifs and thus trigger immune cell activation [1].

CpG ODN2006 primarily targets B cells, inducing cytokine secretion, antibody production and proliferation [2,3,4,5]. Effects of CpG ODN2006 on other TLR9 expressing peripheral blood mononuclear cells (PBMCs) like NK cells, T cells or monocytes are also reported [6,7,8,9] but appear to be rather weak compared to its effects on B cells. Besides its use in in vitro settings, CpG ODN2006 is also under investigation as a drug in diverse clinical applications, however showing limited efficacy [10,11,12,13].

Besides CpG, anti-IgM F(ab’)_2_ fragments (α-IgM antibodies) are used in vivo and in vitro to activate B cells by crosslinking surface IgM molecules [14]. It mimics B cell receptor (BCR) signaling after antigen ligation [2] and causes DNA synthesis, proliferation and apoptosis [15,16,17,18], but only low secretion of IL6, IL8, IL10 and TNF in vitro [2,19].

Recently, we demonstrated that cotreatment with CpG ODN2006 and α-IgM antibodies significantly enhanced the activation of naïve B cells, effectively overcoming their low responsiveness to CpG ODN2006 compared to memory B cells [19]. By combining these two substances, an improved B cell activation including all B cell subsets was achieved, making this combination a also promising tool for potential in vivo applications. However, to the best of our knowledge, no research has been conducted on the impact of either α-IgM alone or in combination with CpG ODN2006 on other PBMC subsets including NK cells, T cells, and monocytes.

To further test the potential of the combined use of CpG ODN2006 and α-IgM antibodies, we investigated their effects on other immune cells within a mixed PBMC culture, as well as in cultures where specific PBMC subtypes were depleted. Understanding the effect of both immune cell stimuli not only on B cells but also across the broader immune cell population provides valuable insights for in vitro and in vivo studies and for the initial evaluation of their effects in future clinical applications.

## 2. Materials and Methods

**Recruitment of donors and blood sampling:** For this study, 34 healthy blood donors (16 women and 18 men) were recruited. None of them had a cancer diagnosis, HBV or HIV infection, or diseases of the lung, lymphatic, cardiovascular, liver or nervous system. All participants provided informed consent. The overall age range was 21–67 years. Women had an average age of 38.4 ± 14.9 (years ± SD), and men had an average age of 38.2 ± 14.2 (years ± SD). Lithium heparin collection tubes were used to take venous blood. The blood was processed within 1 h.

**Reagents used for cell stimulation:** For stimulation, ODN2006 (ODN7909) (InvivoGen, Toulouse, France, #tlrl-2006) in a concentration of 0.195 μM, polyclonal AffiniPure™ F(ab’)_2_ Fragment Goat Anti-Human IgM, Fc_5µ_ fragment specific (Jackson ImmunoResearch, Biozol, Eching, Germany #109-006-129) and monoclonal Mouse F(ab’)2 Anti-Human IgM (µ-chain-specific, clone UHB) Secondary Antibody (MyBioSource, Biozol, Eching, Germany #MBS674003) both in a concentration of 20 μg/mL were used. The monoclonal antibody was used only for results presented in Figure 4.

**Isolation and stimulation of peripheral blood mononuclear cells (PBMCs):** Isolation and stimulation of PBMCs were performed as previously described [19]. Briefly, density gradient centrifugation with Pancoll™ (PAN-Biotech, Fisher Scientific, Schwerte, Germany #P04-60500) was used to isolate PBMCs. Afterward, they were resuspended in fetal bovine serum (FBS, Gibco, Thermo Fisher, Schwerte, Germany #10270-106) containing 10% DMSO (Merck, Darmstadt, Germany #472301-1L-M), frozen, and stored at −180 °C. Cells were thawed for stimulation experiments with B cell medium (RPMI 1640 medium (PAN-Biotech, #P04-16515) with 10% FBS, 1% Penicillin–Streptomycin (Gibco, #15140-122), 1% MEM non-essential amino acids (Gibco, #11140-035), 1% sodium pyruvate (Gibco, #11360-070), 0.1% HEPES (Cytiva, Freiburg im Breisgau, Germany #SH30237.01), and 0.055 mM β-mercaptoethanol (Carl Roth, Karlsruhe, Gemany #4227.3)). After washing, 200,000 cells were seeded in 100 µL of B cell medium into 96-well round bottom plates for two hours resting at 37 °C and 5% CO_2_. The stimulants were added to 100 µL B cell medium.

**Isolation of CD19^+^ B cells:** Dynabeads™ Untouched™ Human B Cells Kit (Thermo Fisher, #11351D) was used to isolate B cells as described in the manufacturer’s guidelines. We assured ≥95% pure B cell fraction via flow cytometry staining of CD19. As described above for mixed PBMCs, 200,000 B cells were plated into 96-well round bottom plates and rested for two hours before the stimulants were added.

**Depletion of CD19^+^ B cells and CD14^+^ monocytes**: To deplete CD19^+^ B cells or CD14^+^ monocytes, respectively, PBMCs were resuspended in depletion buffer (DPBS (Gibco, #14190-094), supplemented with 2% FBS and 2 mM EDTA (Carl Roth, #8043.2), sterile filtered) and incubated with FBS, purified anti-human CD19 antibody (Biolegend, Amsterdam, The Netherlands #302202) or anti-human CD14 antibody (Biolegend, #399202) and mouse IgM (BD Pharmingen™, Heidelberg, Germany #555583). The cell suspension was mixed under rotation for 20 min at 4 °C. Afterward, depletion buffer was added to wash the cells, the cell suspension was centrifuged at 1500 rpm for 5 min at 4 °C and resuspended again in depletion buffer. Cells and washed Dynabeads™ Pan Mouse IgG (Invitrogen, Schwerte, Germany #11041) were mixed under rotation at RT for 15 min and resuspended thoroughly with depletion buffer. The tube was placed into the magnet Dynal MPC^®^-15 (Dynal Biotech, Invitrogen #120.29) for 3 min. The supernatant containing the desired cells was transferred to a new tube and the remaining beads in the new depletion buffer were placed again into the magnet for 3 min. Both obtained supernatants were combined in a fresh tube and again placed into the magnet for 3 min. The obtained cells were counted. The success of the depletion was assessed by CD19 or CD14 staining via flow cytometry. Only samples with the depleted cell subtype making up less than 1% of the total were included for further analysis.

**Flow cytometry staining for detection of α-IgM binding, activation marker expression and apoptosis:** Detailed protocols and antibody panels for analysis via flow cytometry can be found in [19]. 0.2 Mio PBMCs were stained with Zombie UV™ Fixable Viability Dye (1:1000, Biolegend, #423108) and afterward incubated with PBS containing 2% Albumin Fraction V (Carl Roth, #0163.4) for blocking of unspecific binding sites. When mentioned in the figure legend, FCR blocking was performed in parallel to albumin incubation by adding 2.5 µL Human TrueStain FcX™ (1:20, Biolegend, #422302) to the cell suspension. Cells were stained for extracellular markers using an antibody panel for activation and apoptosis markers as well as a panel including (α-)IgM-FITC as described in [19]. After 24 h cultivation of PBMCs, isolated B cells or depleted fractions, the supernatant was taken and stored at −80 °C to perform cytokine analyses at a later timepoint. The stimulated cells were used for measurement of activation markers and stained as described above with the antibody panel for activation markers and apoptosis as described in [19]. The flow cytometer Cytek® Aurora (5 Lasers) was used for analysis.

**Measurement of intracellular cytokines**: PBMCs were stimulated as described above. To prevent secretion of intracellular cytokines Brefeldin A (1:1000, Sigma Aldrich, Merck #B7651-5MG) was added to the cells at the start of cultivation or after 18 h of initial stimulation. The protocol and antibody panels used for staining PBMCs for intracellular cytokines are described in detail in [19]. After staining of dead cells by Zombie UV™ Fixable Viability Dye, unspecific binding sides were blocked with PBS containing 2% Albumin Fraction V and extracellular markers were stained as in [19]. A fixation and permeabilization step with 2% paraformaldehyde and 1:10 diluted FACS™ Permeabilizing Solution 2 followed, each for 10 min at RT in the dark. Afterward, intracellular cytokines were stained using the antibody panel as described in [19]. After staining, cells were measured at the flow cytometer Cytek^®^ Aurora (5 Lasers).

**Measurement of secreted cytokines:** Measurements were performed as previously described [19]. Briefly, to measure cytokines that were released by mixed PBMCs, isolated B cells or B cells- and monocytes-depleted fractions into the supernatant during stimulation, samples were processed via LEGENDplex™ Human Inflammation Panel 1 (Biolegend, #740809) multiplex assay as described in the kit guidelines but with threefold reduced volume of each component. For IL6, IL8 and MCP1, due to the very high concentrations, ELISAs were performed using the following kits: ELISA MAX™ Standard Set Human IL-6 (Biolegend, #430501), ELISA MAX™ Standard Set Human IL-8 (Biolegend, #431501), ELISA MAX™ Standard Set Human MCP-1/CCL2 (Biolegend, #438807). Here, the appropriate dilutions of the samples were determined in prior experiments and all volumes were scaled down by a factor of two.

**Software and data analysis:** FlowJo (Version 10.3) was used for analyzing flow cytometric data, including t-SNE analysis. To perform t-SNE analysis of monocytes stained for intracellular cytokines, the following markers were included: CD14, IL1β, TNF, MCP1, IL6, IL8, IL10. LEGENDplex™ QOGNIT (legendplex.qognit.com (accessed on 21 December 2022)) was used for analysis of multiplex data. Prism 9 (GraphPad Software, v 9.2.0) was used for analysis of ELISA data.

**Statistical analysis and illustration:** For statistical analyses and illustration, the program Prism 9 software (GraphPad Software, v 9.2.0) was used. More details regarding statistical analyses can be found in the figure legends. ELISA and multiplex measurements were performed in duplicates while flow cytometry experiments as single measurements.

## 3. Results

### 3.1. α-IgM Acted as a Strong, but Not B Cell-Specific Inducer of Cytokine Secretion Compared to CpG ODN2006 in Mixed PBMCs

Simultaneous use of CpG ODN2006 and α-IgM has been shown to be a beneficial tool to achieve optimal B cell stimulation [19], but its effects on other PBMC subtypes are rarely studied yet. The secretion of cytokines by many immune cells represents an important key player for inducing further diverse immune reactions, both for in vitro use as well as in clinical applications. Thus, we measured the cytokine profile using the supernatant of mixed PBMCs stimulated with combined CpG ODN2006 and α-IgM.

Interestingly, we observed a strong increase, especially for MCP1, IL6 and IL8, a moderate induction of IL1β, TNF and IL10 secretion as well as a low increase in IFNα2, IFNγ, IL12p70, IL18, IL23 and IL33 (Figure 1A, Appendix A). When analyzing the secretion induced by each stimulant alone, we found that CpG ODN2006 led to only a low increase in cytokine levels, while a similar pattern was obtained for α-IgM alone compared to the combination of CpG ODN2006 and α-IgM (Figure 1B, Appendix A). These findings suggest that α-IgM is the main inducer of the strong cytokine production observed in mixed PBMCs.

In a previous study, we confirmed the effective activation of B cells by CpG ODN2006 and α-IgM especially by their combined use [19]. Thus, we investigated whether B cells are responsible for the secretion of these cytokines by performing intracellular cytokine staining for the six cytokines produced in the highest amounts, namely IL1β, TNF, MCP1, IL6, IL8 and IL10. Our results revealed a robust induction of TNF, IL6 and IL8 production in B cells within the first 6 h of stimulation mainly by CpG ODN2006 in the presence of α-IgM, which was still visible after 24 h (Figure 1C), matching our previous findings [19]. In contrast, IL10-expressing B cells were detectable only after 24 h of TLR9 stimulation and BCR crosslinking. Interestingly, B cells did not show positive intracellular staining for MCP1 or IL1β with any of the applied stimuli. These findings match with the cytokine profile that was obtained from the supernatant of isolated B cells after exposure to both immune cell stimulants (Figure 1D). Notably, after CpG ODN2006 and α-IgM stimulation, almost all measured cytokines—except IL10—were found in higher concentrations when mixed PBMCs were stimulated compared to isolated B cells (Appendix A). This suggests that B cells are not the sole source of these soluble factors. In line, we could not detect IL1β or MCP1 production or secretion by B cells (Figure 1C,D). Thus, we hypothesized that another PBMC subset is also involved in cytokine secretion.

### 3.2. α-IgM Induced Strong Cytokine Production by Monocytes

To identify the source of highly released cytokines after α-IgM exposure in mixed PBMCs, we stained for intracellular IL1β, TNF, MCP1, IL6, IL8 and IL10 in all PBMC subsets after 6 h and 24 h of stimulation. Using t-distributed stochastic neighbor embedding (t-SNE) analysis, we could show a correlation between high cytokine expression and CD14^+^ monocytes (Figure 2A). In line, t-SNE plots after gating on CD14^+^ monocytes indicate strong changes in cytokine production upon α-IgM stimulation compared to unstimulated monocytes (Figure 2B). After 6 h, a high number of IL1β-, TNF-, IL6- and IL8-expressing monocytes was found after α-IgM but not CpG ODN2006 stimulation (Figure 2C). These increased expression levels were still detectable but strongly declined or even entirely lost after 24 h. Rapid cytokine secretion is also confirmed by the time course measurement of the highly secreted cytokines IL6 and IL8 (Appendix A). In contrast, MCP1 expression was initiated at a later time point shown in intracellular cytokine staining (Figure 2C) and measurement of MCP1 in the supernatant over time (Appendix A). Thus, we assume an indirect induction of MCP1 production, maybe indirectly by the earlier secreted cytokines. Of note, T cells and NK cells did not show any biologically relevant changes in cytokine production (Appendix A). The depletion of CD14^+^ monocytes from mixed PBMCs (Appendix A) supported that most released cytokines are monocyte-derived since secretion of IL1β, TNF, MCP1, IL6, IL8 and IL10 in the α-IgM-stimulated culture without monocytes was strongly reduced compared to complete PBMCs (Figure 2D,E). Thus, we identified the monocytes to be the main source of secreted cytokines after α-IgM stimulation.

### 3.3. CpG ODN2006 Did Not Induce Cytokine Secretion from Monocytes

We observed that CpGODN2006 induced a much lower number of cytokine-expressing monocytes compared to α-IgM. Consistent with this, t-SNE analysis of intracellular cytokines in CpG ODN2006-stimulated monocytes showed very similar clustering compared to unstimulated cells (Appendix A), indicating no change in cytokine production. In contrast, the supernatant from mixed PBMCs showed increased levels of TNF, IL6, IL8 and IL10 after CpG ODN2006 stimulation (Appendix A). The increase in TNF, IL6 and IL10 was comparable even after depletion of monocytes, indicating that these cytokines were produced mainly by B cells and not by monocytes after CpG ODN2006 exposure. For IL8, its CpG ODN2006-induced secretion by mixed PBMCs was significantly reduced when CD14^+^ monocytes were depleted (Appendix A). Since intracellular IL8 staining in monocytes was negative (Figure 2B), we assume that IL8 production by B cells after CpG ODN2006 stimulation is supported by monocytes. With these findings, we conclude that CpG ODN2006—in comparison to α-IgM—mainly activates the cytokine production of B cells but not of monocytes.

### 3.4. α-IgM Showed Direct Monocyte Binding

Next, we investigated whether the induction of cytokine production in monocytes is caused indirectly by interplay with activated B cells. Thus, we depleted CD19^+^ B cells from PBMCs (Appendix A) and stimulated the B cell-depleted fraction with α-IgM under consistent conditions. The lack of B cells did not affect the secretion of cytokines—which are mainly monocyte-derived—as mixed PBMCs and the B cell-depleted fraction show comparable cytokine levels after α-IgM exposure (Figure 3A,B). We conclude that the action of α-IgM on monocytes and their following cytokine secretion is B cell-independent. Additionally, these data confirm, that the antibody fragment alone is only a weak inducer of cytokine production in B cells as already observed in isolated B cells (Figure 1) and described previously [19].

To test the hypothesis of direct binding of α-IgM to monocytes, we incubated monocytes with FITC-conjugated α-IgM and tracked binding via flow cytometry. Although we expected B cells to be the only subset expressing IgM, we detected direct binding of the antibody fragment to monocytes—though at a significantly lower intensity compared to its binding to B cells (Figure 3C,D). Based on new findings on the immunoglobulin expression of monocytes [20,21], we investigated whether monocytes express IgM by using a commercial α-IgM-Ab. However, we did not detect IgM^+^ monocytes (Figure 3E). Furthermore, α-IgM binding to monocytes could not be prevented by Fc receptor blocking (Figure 3F), as expected due to the absence of the Fc portion on the fragment. Thus, the binding mode of α-IgM is not clarified yet.

To determine if cross-reactivity of the polyclonal antibody fragment with other antigens was responsible for monocyte activation, we investigated whether a monoclonal α-IgM-F(ab’)_2_ fragment also induces an increase in cytokine production. We observed enhanced levels of cytokines within monocytes (Figure 4A) and in the supernatant of mixed PBMCs (Figure 4B).

Although the binding mechanism of the antibody fragment to monocytes requires further investigation, our findings suggest that α-IgM may influence monocytes due to direct binding.

### 3.5. CpG ODN2006 and α-IgM Increased Activation Marker Expression on Monocytes

We further investigated whether CpG ODN2006 and α-IgM influence apoptosis and activation marker expression on monocytes after 6 h or 24 h of stimulation. Interestingly, CpG ODN2006 stimulation resulted in increased CD86, HLA-DR and CD69 expression on monocytes (Figure 5A–C), without inducing apoptosis (Figure 5D). This suggests an activating role of CpG ODN2006 towards monocytes, albeit without stimulating their cytokine production (Figure 2C). Stimulation with α-IgM for 24 h resulted in a slight induction of apoptosis (Figure 5H), possibly explaining the reduced cytokine production after 24 h compared to 6 h (Figure 2A–C). The strong cytokine production after 6 h also correlates with increased CD86 and HLA-DR expression levels by polyclonal α-IgM (Figure 5E,F), which was confirmed with monoclonal α-IgM (Appendix A). This α-IgM-induced increase returned to baseline levels after 24 h (Figure 5E,F), while CD69 expression did not show a significant change (Figure 5G). Similar effects on activation marker expression induced by CpG ODN2006 and α-IgM could be observed after depletion of B cells from the mixed PBMC culture (Appendix A), suggesting that the activation of monocytes occurs independently of B cells.

### 3.6. Strong Cytokine Secretion of Monocytes Did Not Affect B Cell Activation Markers

Since we observed strong cytokine production induced by α-IgM in monocytes, we investigated whether this has an additional stimulatory effect on co-cultured B cells. To address this, we conducted two approaches: first, using isolated B cells, and second, depleting CD14^+^ monocytes from mixed PBMCs. We then compared the activation marker expression on B cells under these three conditions. Overall, comparable expression of activation markers CD69, CD71, CD86 and HLA-DR after stimulation with CpG ODN2006 and α-IgM and their combination was observed for B cells cultured in complete or monocyte-depleted PBMCs, as well as for isolated B cells (Figure 6). In conclusion, activated monocytes do not affect the activation markers of B cells via CpG ODN2006, α-IgM and their combination.

## 4. Discussion

While CpG ODN2006 and α-IgM-Abs are conventionally described for their synergistic effects in stimulating B cells, leading to increased expression of activation markers and cytokine secretion [19,22], we investigated their impact on mixed PBMCs to better simulate an in vivo scenario. Of note, as TLR expression and sequence-recognition pattern are not comparable between human and animals, in vivo models are not suitable for investigating the effects of CpG [23]. By using mixed PBMCs, we were able to examine whether there are (1) direct or indirect B cell-driven effects of CpG ODN2006 and α-IgM on other immune cells and (2) indirect influences of other PBMC subsets on B cells.

It is reported that monocytes express TLR9 [24]. In line, we could demonstrate that CpG ODN2006 increased CD86, HLA-DR and CD69 expression on monocytes but without inducing strong cytokine secretion. Thus, the effects of CpG ODN2006 on monocytes appear to be weak in our settings. In contrast, we showed strongly increased secretion of monocytic cytokines by α-IgM, which has not been described yet. This strong cytokine production occurred mainly within the first 6 h of stimulation while a decline was observed again after 24 h. We assume that the fast decline in the production of these cytokines is due to the induction of apoptosis triggered by strong activation.

The observation of strong cytokine production at early time points suggested a direct effect of α-IgM on monocytes. Flow cytometry revealed direct binding of the antibody fragment to monocytes. However, we excluded binding via IgM and FcRs, as monocyte IgM staining was negative and α-IgM binding to monocytes could not be prevented by Fc receptor blocking. Therefore, the exact mechanism of interaction between the antibody fragment and monocytes is unclear. It is conceivable that the fragment binds to IgM-like structures on monocytes. Additionally, immune complexes could form, potentially via aggregation of the antibody fragments, and interact with cell surface receptors, consequently resulting in an activation of monocytes. Further research is needed to clarify the specific mechanisms and binding mode.

To confirm the direct effect of α-IgM on monocytes, we isolated and cultured monocytes for 24 h. Unfortunately, due to their short lifespan [25] and the freezing/thawing process, we observed that most monocytes died without any additional stimulus within 24 h, supporting the necessity of the presence of other PBMC subtypes for maintaining their viability. Nevertheless, we did not use any additional substances to keep the monocytes alive, as it would preclude a direct comparison of the results of isolated monocytes and monocytes in mixed PBMCs. Instead, we introduced B cell depletions to eliminate possible secondary effects induced by activated B cells. We observed a comparable cytokine secretion profile in B cell-depleted cell cultures compared to mixed PBMC. Thus, we excluded the possibility of indirect activation of monocytes via B cells.

Few studies describe the effects of CpG ODN2006 on T cells and NK cells [6,7,8,9]. For instance, Papagno et al. have shown that CpG ODN2006 is less effective in supporting T cell responses compared to other adjuvants [26], which was supported in our study, where no CpG ODN2006-induced cytokine secretion of T cells and NK cells was observed. This strengthens the consideration of CpG ODN2006 as a B cell stimulant. Also, α-IgM did not lead to an increase in cytokine production of T cells and NK cells as it did for monocytes.

It is well described that plasmacytoid dendritic cells (pDCs) are also activated by class B CpG leading mainly to secretion of IFNα [27,28]. Generally, the percentage of pDCs within mixed PBMCs is low. Furthermore, the amount of secreted IFNα2, which is a typical cytokine secreted by pDCs after TLR9 stimulation [29], in the supernatant of mixed PBMCs was very low after stimulation with CpG ODN2006 and α-IgM and likely lacks biological relevance. For these reasons, the influence of pDCs in the setting of mixed PBMCs was neglected.

Since α-IgM stimulation led to strong production of cytokines by stimulated monocytes after already early time points, we tested for monocyte-induced effects on B cells. We could prove that the lack of monocytic cytokines did not affect B cell activation markers within 24 h of stimulation. Since we obtained a plateau of activation markers and cytokines already after 18 h reaching out 24 h, we focused on these short-term effects and later time points were not considered in this study. However, indirect effects after a longer incubation period cannot be excluded.

In the clinical context, CpG ODN2006 lacks convincing efficacy in first clinical trials including anti-cancer treatment and vaccinations [6,11,12,13,30]. To test for a better induction of desired immune responses we combined CpG ODN2006 with α-IgM antibodies, which also gained attention in other clinical applications [31]. In this study, we observed high levels of secreted cytokines produced by monocytes after α-IgM exposure indicating a cytokine storm, which makes it a rather unfavorable drug for medical use. When using α-IgM as a potential combinatorial stimulus to CpG ODN2006 for clinical applications, it would be essential to find a safe dose that still activates B cells and monocytes efficiently but prevents such a strong cytokine secretion that we have seen in vitro. Overall, these findings emphasize the importance of also considering the immunological responses in a setting of mixed PBMCs.

## 5. Conclusions

Combined use of α-IgM together with CpG ODN2006 can activate B cells very potently. In this study, we showed that α-IgM also activated monocytes besides B cells. This was seen in increased activation marker expression and especially strongly induced cytokine production due to direct binding and in a B cell-independent manner. Of note, these activated monocytes did not affect B cell activation markers. Thus, the use of CpG ODN2006 in combination with α-IgM remains an effective strategy for promoting optimal B cell activation. Nevertheless, it should be considered that the released cytokines after stimulation of mixed PBMCs in vitro are mainly monocyte-derived. Importantly, for testing the combined use of CpG ODN2006 and α-IgM in clinical applications, a strong focus should be set on possible unwanted high cytokine secretion by monocytes as indicated by the findings within this in vitro study.

## Figures and Tables

**Figure 1 cells-13-01822-f001:**
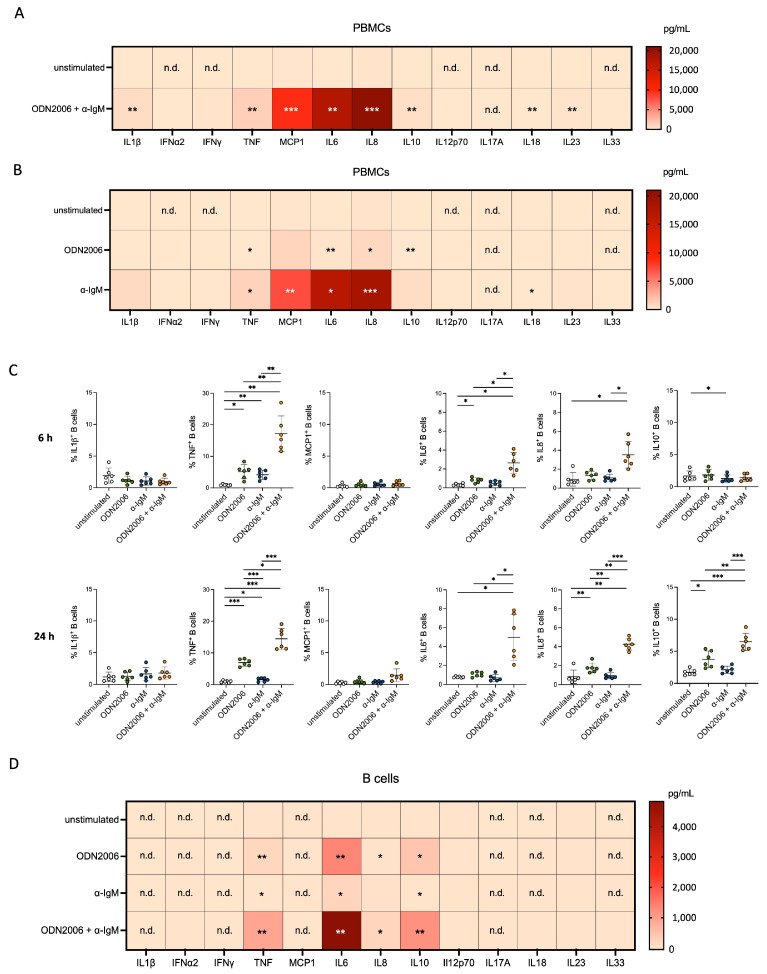
Strong cytokine release after α-IgM exposure in mixed PBMCs mainly not attributable to B cells. (**A**,**B**) Heat map diagram of secreted IL1*β*, IFN*α*2, IFN*γ*, TNF, IL10, IL12p70, IL17A, IL18, IL23, IL33 (measured via Legendplex, *n* = 7) and MCP1, IL6, IL8 (measured via ELISA, *n* = 9–11) in supernatant of mixed PBMCs 24 h after treatment with ODN2006 + α-IgM (**A**) or both individually (**B**). (**C**) Intracellular cytokine staining for IL1*β*, TNF, MCP1, IL6, IL8 and IL10 in B cells after ODN2006, α-IgM or ODN2006 + α-IgM stimulation for 6 h (**top**) and 24 h (**bottom**) via flow cytometry staining (*n* = 6). (**D**) Heat map diagram of cytokines and chemokines released in supernatant of isolated B cells after 24 h of stimulation with ODN2006, α-IgM or their combination obtained by Legendplex Human Inflammatory Panel 1 (*n* = 5). N.d.: not detectable. No statistical test was performed for cytokines whose level was not detectable in unstimulated sample. Paired *t* test was used for the comparison of unstimulated vs. stimulated conditions (**A**). One-Way ANOVA with Geisser-Greenhouse correction and Tukey’s multiple comparisons test was used for analyzing data including more than two stimulation conditions (**A**–**D**); * *p* ≤ 0.05, ** *p* ≤ 0.01, *** *p* ≤ 0.001. n.d. = not detectable.

**Figure 2 cells-13-01822-f002:**
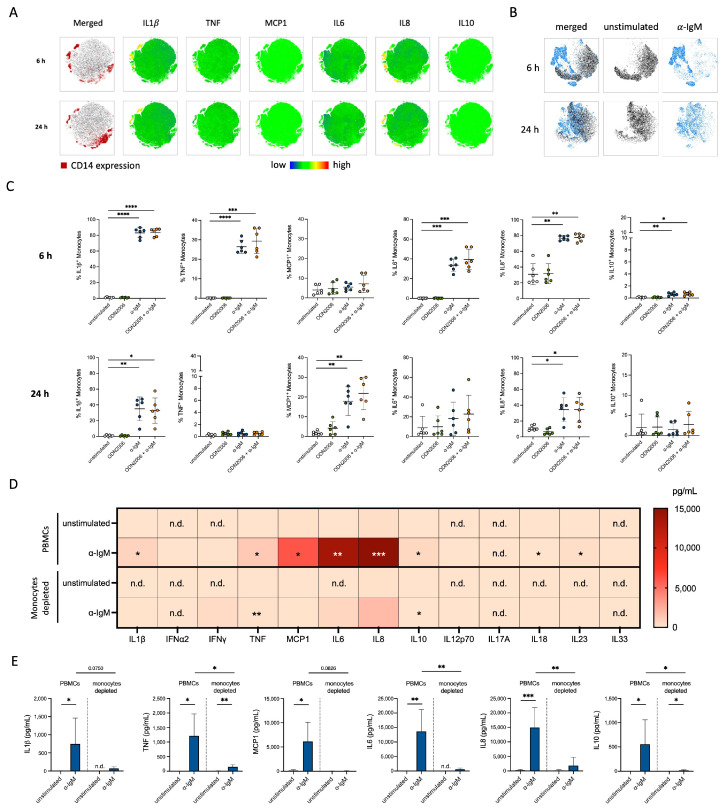
α-IgM caused strong cytokine production in monocytes. (**A**) t-SNE plots from mixed PBMCs (monocytes highlighted in red) that were stimulated with α-IgM for 6 h (top) and 24 h (bottom) as well as t-SNE plots displayed as a heatmap for IL1*β*, TNF, MCP1, IL6, IL8, IL10 (*n* = 6). (**B**) t-SNE plots from samples stained for intracellular cytokines that were unstimulated (black) or stimulated with α-IgM (blue) for 6 and 24 h after gating on CD14^+^ monocytes (*n* = 6). (**C**) Staining of intracellular IL1*β*, TNF, MCP1, IL6, IL8 and IL10 in monocytes after ODN2006, α-IgM or ODN2006 + α-IgM stimulation for 6 h (top) and 24 h (bottom) via flow cytometry (*n* = 6). (**D**) Heat map diagram of secreted IL1*β*, IFN*α*2, IFN*γ*, TNF, IL10, IL12p70, IL17A, IL18, IL23, IL33 (measured via Legendplex, *n* = 6) and MCP1, IL6, IL8 (measured via ELISA, *n* = 6–8) in supernatant of mixed PBMCs (top) or monocyte-depleted fraction (bottom) after 24 h of stimulation with α-IgM. (**E**) Secreted IL1*β*, TNF, MCP1, IL6, IL8 and IL10 in supernatant of mixed PBMCs and monocyte depleted fraction after α-IgM stimulation for 24 h measured with Legendplex for IL1*β*, TNF and IL10 (*n* = 6; statistical asterisks above bars refer to the unstimulated sample of the same cells) and ELISA for MCP1, IL6 and IL8 (*n* = 7–8; statistical asterisks above bars refer to the unstimulated sample of the same cells). n.d.: not detectable. No statistical test was performed for cytokines whose level was not detectable in unstimulated sample. One-Way ANOVA or mixed-effects analysis with Geisser–Greenhouse correction and Tukey’s multiple comparisons test was used for analyzing data including more than two stimulation conditions (**C**). Paired *t* test was used for the comparison of unstimulated vs. stimulated conditions within the same cell type as well as for comparing data from PBMCs with data from monocytes depleted fraction (**D**,**E**); * *p* ≤ 0.05, ** *p* ≤ 0.01, *** *p* ≤ 0.001, **** *p* ≤ 0.0001.

**Figure 3 cells-13-01822-f003:**
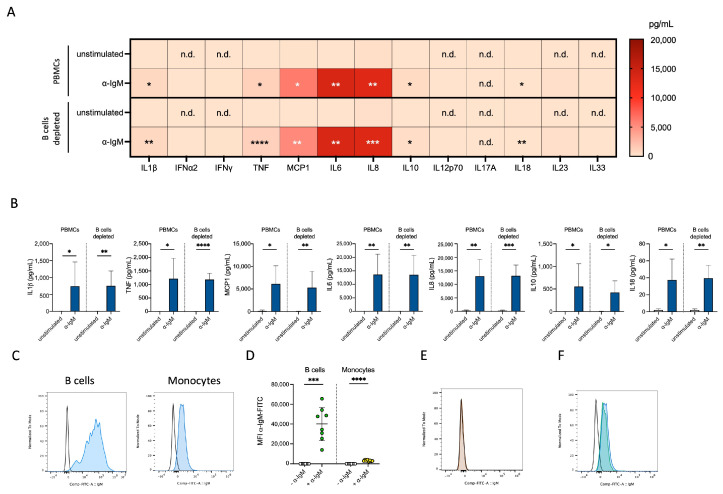
α-IgM acted directly on monocytes. (**A**,**B**) Heat map (**A**) and bar diagrams (**B**) of secreted IL1*β*, IFN*α*2, IFN*γ*, TNF, IL10, IL12p70, IL17A, IL18, IL23, IL33 (measured via Legendplex) and MCP1, IL6, IL8 (measured via ELISA) in supernatant of mixed PBMCs (top or left) or B-cell-depleted fraction (bottom or right) after 24 h of stimulation without or with α-IgM (*n* = 6–8, statistical asterisks in heat map refer to the unstimulated sample of the same cells). (**C**) Representative histograms of unstimulated B cells and monocytes using α-IgM-FITC (*n* = 8). Empty histograms: negative controls. (**D**) Quantification of MFI of unstimulated B cells, monocytes using α-IgM-FITC (*n* = 8). (**E**,**F**) Representative histograms showing unstimulated monocytes stained with commercially available IgM-FITC (**E**) and α-IgM-FITC without (blue) or with (green) FCR blocking (**F**). Empty histograms: negative controls. n.d.: not detectable. No statistical test was performed for cytokines whose level was not detectable in unstimulated sample. Paired *t* test was used for the comparison of unstimulated vs. stimulated conditions within the same cell type as well as for comparing data from PBMCs with data from B-cell-depleted fraction (**A**,**B**,**D**); * *p* ≤ 0.05, ** *p* ≤ 0.01, *** *p* ≤ 0.001, **** *p* ≤ 0.0001.

**Figure 4 cells-13-01822-f004:**
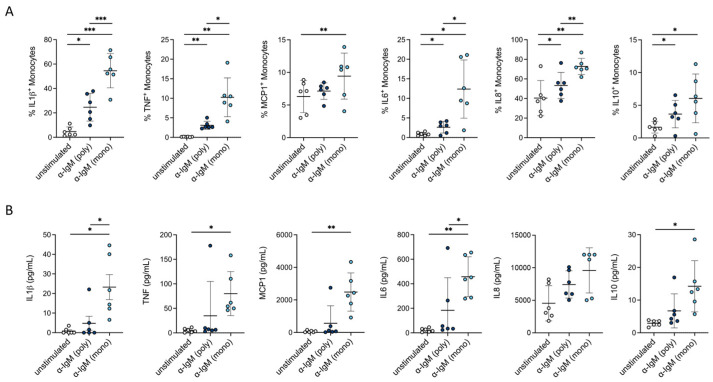
Monoclonal α-IgM also led to an upregulated cytokine production in monocytes. (**A**) Intracellular cytokine staining for IL1*β*, TNF, MCP1, IL6, IL8 and IL10 in monocytes after stimulation with monoclonal and polyclonal α-IgM for 6 h via flow cytometry staining (*n* = 6). (**B**) Secreted IL1*β*, TNF, MCP1, IL6, IL8 and IL10 (measured via Legendplex, *n* = 6) in supernatant of mixed PBMCs after stimulation with monoclonal and polyclonal α-IgM for 24 h. One-Way ANOVA with Geisser–Greenhouse correction and Tukey’s multiple comparisons test was used for analyzing data including more than two conditions (**A**,**B**); * *p* ≤ 0.05, ** *p* ≤ 0.01, *** *p* ≤ 0.001.

**Figure 5 cells-13-01822-f005:**
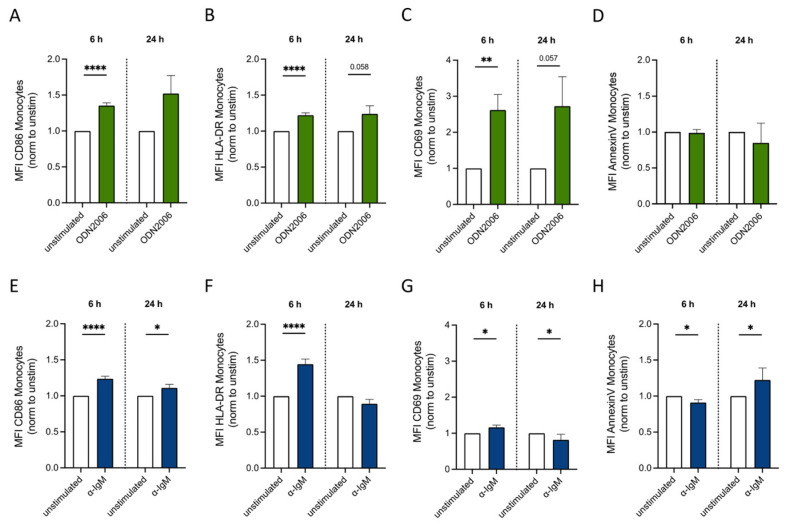
CpG ODN2006 and α-IgM affected activation marker expression on monocytes. (**A**–**D**) Expression of CD86 (**A**), HLA-DR (**B**), CD69 (**C**) and AnnexinV (**D**) on monocytes after 6 h and 24 h of ODN2006 stimulation in culture of mixed PBMCs (*n* = 12). (**E**–**H**) Expression of CD86 (**E**), HLA-DR (**F**), CD69 (**G**) and AnnexinV (**H**) on monocytes after 6 h and 24 h of α-IgM stimulation in culture of mixed PBMCs (*n* = 12). Paired *t* test was used to compare two conditions; * *p* ≤ 0.05, ** *p* ≤ 0.01, **** *p* ≤ 0.0001.

**Figure 6 cells-13-01822-f006:**
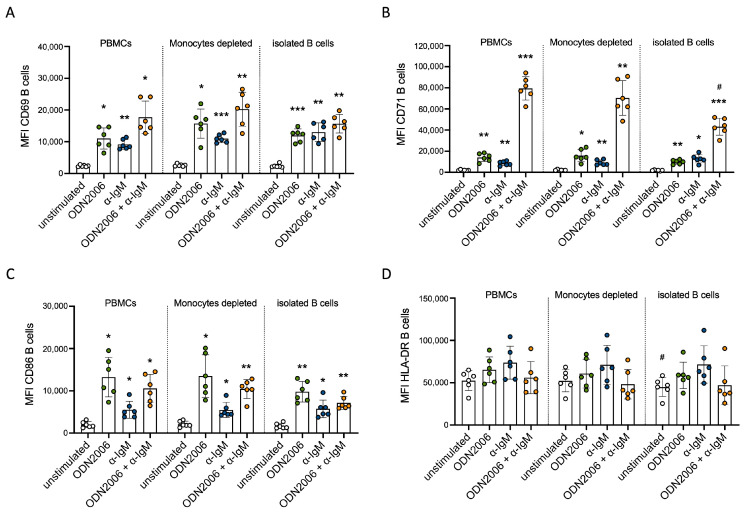
Activated monocytes did not influence B cell activation. (**A**) Comparison of expression of CD69 (**A**), CD71 (**B**), CD86 (**C**) and HLA-DR (**D**) on B cells stimulated with ODN2006, α-IgM or their combination in either a culture of mixed PBMCs, monocyte-depleted fraction or isolated B cells for 24 h (*n* = 6); statistical asterisks above bars refer to the unstimulated sample of the same cells). One-Way ANOVA with Geisser–Greenhouse correction and Tukey’s multiple comparisons test was used for analyzing data; * *p* ≤ 0.05 ** *p* ≤ 0.01, *** *p* ≤ 0.001, ^#^ *p* ≤ 0.05 (refers to PBMC sample under the same conditions).

## Data Availability

On reasonable request, the corresponding author will provide access to the data used and/or analyzed in this study.

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
