# Peer review of "Simultaneous Stimulation of Peripheral Blood Mononuclear Cells with CpG ODN2006 and α-IgM Antibodies Leads to Strong Immune Responses in Monocytes Independent of B Cell Activation"

_cells, 2024, doi:10.3390/cells13221822_

Round 1
Reviewer 1 Report
Comments and Suggestions for Authors
The authors describe an investigation of the effects of α-IgM, CpG ODN2006, and a combination of both on PBMCs and peripheral immune subsets. As the authors have highlighted, the combination is very successful in activating B cells in isolation, but the effects on a mixed population are an important and missing aspect of the current literature.
The experimental design described is suitable, no obvious ommissions have been made, and the conclusions drawn are supported by the observations.
This is an important addition to the current knowledge as it highlights the potential for confounding effects should treatment be made in a mixed PBMC population.
Author Response
The authors describe an investigation of the effects of α-IgM, CpG ODN2006, and a combination of both on PBMCs and peripheral immune subsets. As the authors have highlighted, the combination is very successful in activating B cells in isolation, but the effects on a mixed population are an important and missing aspect of the current literature.
The experimental design described is suitable, no obvious ommissions have been made, and the conclusions drawn are supported by the observations.
This is an important addition to the current knowledge as it highlights the potential for confounding effects should treatment be made in a mixed PBMC population.
Reply: thank you very much for your positive feedback!
Reviewer 2 Report
Comments and Suggestions for Authors
This is an informative manuscript on the differential influences of CpG ODNs, F(ab)2 anti-IgM, (a-IgM antibodies) and their combination on the cells in human PBMC preparations. Removing B cells or monocytes from the PBMC preparation further adds to some new information on some direct stimulatory effects of a-IgM antibodies on monocytes. There are some concerns about wording and claims made, and these will be specifically noted with some suggested changes (in bold and underlined).
Line 49: “ cotreatment with CpG ODNs and a-IgM antibodies” Since it states cotreatment, “and CpG ODN2006” should be added
Line 51: “to their memory counterparts” to memory B cells This assumes their memory counterparts is meaning memory B cells.
Line 52: “ including all B cell subsets” It may be more accurate to have additional B cell subsets rather than all subsets which assumes we have differentially identified all subsets.
Reagents: Two (Fab)2 ant-human IgM were described with one being a polyclonal and one a monoclonal. In results and discussion, the anti-IgM is referred to as a-IgM antibodies. It needs to be made clearer when the polyclonal or monoclonal were used. Furthermore, the epitope specificity of the monoclonal should be provided. The source of the α-IgM antibodies needs to be clarified as well as when the polyclonal or monoclonal antibody is used as was shown in Figure 4. Do the authors have an explanation for why the monoclonal is more able to stimulate monocytes (Fig. 4) ?
The “B cell medium” contains 1.918 ml of b-mercaptoethanol (b-ME). The final concentration of the b-ME in the medium should be provided. Additionally, it is unclear why b-ME is used; b-ME should be needed only for in vitro analysis of mouse immune cells since mice have lower glutathione levels than humans. Furthermore, the reducing agent may be affecting some disulfides of cells or proteins, e.g., the Fc of Ig or even the Fab as it's known to affect IgG4 Fab exchange. Reducing agents or known to affect Ig structures and surface receptors. Since the culturing is done at atmospheric pO2 and not physiological pO2 (in vivo blood pO2 is 47-62% that of atmospheric air), the b-ME might actually form disulfides with R-SH of cells changing protein structures.
When B cells were present it would have been useful to know the amount of IgM released which would have led to immune complexes if some of the α-IgM antibodies were still available in the medium.
Line 343: “ produced exclusively mainly by B cells’
Line 401: “unstimulated B cells and monocytes using”
Line 450-451: without stimulating their cytokine production (Fig. 2C).
Line 563: “we excluded binding via IgM and FcRs” It would help here to remind readers how you made this exclusion.
Line 595: “we could prove can suggest” add at the end of the sentence since…
Author Response
This is an informative manuscript on the differential influences of CpG ODNs, F(ab)2 anti-IgM, (a-IgM antibodies) and their combination on the cells in human PBMC preparations. Removing B cells or monocytes from the PBMC preparation further adds to some new information on some direct stimulatory effects of a-IgM antibodies on monocytes. There are some concerns about wording and claims made, and these will be specifically noted with some suggested changes (in bold and underlined).
Reply: Thank you very much for your positive feedback.
Line 49: “ cotreatment with CpG ODNs and a-IgM antibodies” Since it states cotreatment, “and CpG ODN2006” should be added
Line 51: “to their memory counterparts” to memory B cells This assumes their memory counterparts is meaning memory B cells.
Line 52: “ including all B cell subsets” It may be more accurate to have additional B cell subsets rather than all subsets which assumes we have differentially identified all subsets.
Reply: The suggested changes are integrated in the revised version of the manuscript.
Reagents: Two (Fab)2 ant-human IgM were described with one being a polyclonal and one a monoclonal. In results and discussion, the anti-IgM is referred to as a-IgM antibodies. It needs to be made clearer when the polyclonal or monoclonal were used. Furthermore, the epitope specificity of the monoclonal should be provided. The source of the α-IgM antibodies needs to be clarified as well as when the polyclonal or monoclonal antibody is used as was shown in Figure 4. Do the authors have an explanation for why the monoclonal is more able to stimulate monocytes (Fig. 4) ?
Reply:Thank you for this comment. We specified in the section “reagents” that the monoclonal antibody was used only for results presented in figure 4. The sources (mouse) and epitope specificity and clone number ((µ chain specific, clone UHB) are included in the revised manuscript.
The “B cell medium” contains 1.918 ml of b-mercaptoethanol (b-ME). The final concentration of the b-ME in the medium should be provided. Additionally, it is unclear why b-ME is used; b-ME should be needed only for in vitro analysis of mouse immune cells since mice have lower glutathione levels than humans. Furthermore, the reducing agent may be affecting some disulfides of cells or proteins, e.g., the Fc of Ig or even the Fab as it's known to affect IgG4 Fab exchange. Reducing agents or known to affect Ig structures and surface receptors. Since the culturing is done at atmospheric pO2 and not physiological pO2 (in vivo blood pO2 is 47-62% that of atmospheric air), the b-ME might actually form disulfides with R-SH of cells changing protein structures.
Reply: Thank you for poiting out this issue. We added the final concentration of b-ME used. We agree that b-ME is not necessarly required for human B cells, but it supports cells viability, and it is essential in serum-free culture conditions and in mouse B cell culture. To keep our protocols comparable and applicable to many conditions in the lab, we used b-ME also in the experiments presented here. The hypothesis that b-ME could affect the antibody structure and eventually its binding, it is very intriguing. However, IgM binding is more stable than IgG, therefore this hypothesis should be carefully investigated in a future step. Considering the large amount of studies conducted with b-ME-containing culture medium (especially but not exclusively in mouse model), we think that the presence of b-ME in the culture medium does not affect the relevance of the presented results.
When B cells were present it would have been useful to know the amount of IgM released which would have led to immune complexes if some of the α-IgM antibodies were still available in the medium.
Reply: This is a good suggestion and we will consider this point in future studies.
Line 343: “ produced exclusively mainly by B cells’
Line 401: “unstimulated B cells and monocytes using”
Line 450-451: without stimulating their cytokine production (Fig. 2C).
Line 563: “we excluded binding via IgM and FcRs” It would help here to remind readers how you made this exclusion.
Line 595: “we could prove can suggest” add at the end of the sentence since…
Reply: We made the changes suggested, as highlighted in the revised version of the manuscript.
Reviewer 3 Report
Comments and Suggestions for Authors
This excellently written and presented paper deals with the effects that CpG ODN2006 and anti-IgM antibodies have on the in vitro stimulation and mainly on the cytokine secretion of human B cells and monocytes. It includes a good number of well designed and very informative figures. It extends and deepens on previous observations of the authors.
Through a series of concatenated experiments, the authors convincingly demonstrate that:
1. CpG ODN2006 – in comparison to anti-IgM – mainly activates the cytokine production of B cells but not of monocytes.
2. The anti-IgM antibodies bind to -by a way not identified yet- and actívate monocytes independently of B cells.
As a conclusion, the high cytokine secretion by monocytes upon stmulation with anti-IgM antibodies to enhance CpG ODN2006’s efficacy on B cells, should be considered as a rather unwanted effect, particularly in clinical settings, such as in an anti-cancer regime.
Author Response
This excellently written and presented paper deals with the effects that CpG ODN2006 and anti-IgM antibodies have on the in vitro stimulation and mainly on the cytokine secretion of human B cells and monocytes. It includes a good number of well designed and very informative figures. It extends and deepens on previous observations of the authors.
Through a series of concatenated experiments, the authors convincingly demonstrate that:
- CpG ODN2006 – in comparison to anti-IgM – mainly activates the cytokine production of B cells but not of monocytes.
- The anti-IgM antibodies bind to -by a way not identified yet- and actívate monocytes independently of B cells.
As a conclusion, the high cytokine secretion by monocytes upon stmulation with anti-IgM antibodies to enhance CpG ODN2006’s efficacy on B cells, should be considered as a rather unwanted effect, particularly in clinical settings, such as in an anti-cancer regime.
Reply: thank you very much for this very positive feedback!
Reviewer 4 Report
Comments and Suggestions for Authors
Summary
In this study, the combined effects of CpG ODN2006 and α-IgM antibodies, typically used for B cell activation, were examined on peripheral blood mononuclear cells (PBMCs). While previous research has focused on optimizing B cell activation using these agents, this work extends the analysis to include their impact on other immune cell types. Using flow cytometry and cytokine assays, the study identified that α-IgM, though effective in stimulating monocytes as evidenced by increased CD86 and HLA-DR expression, led to elevated monocyte-derived cytokine production. This activation, independent of B cells, suggests a potential drawback for clinical applications due to monocyte-driven side effects, even though it does not impede B cell activation. The authors conclude that while α-IgM can potentiate CpG ODN2006 in B cells, it may also cause undesirable monocyte activation, necessitating caution when considering this combination for therapeutic use.
Overall comment
The experimental design in this study is well-conceived. The statistical analyses applied are appropriate for the data. The data presentation is clear and well-organized, with detailed descriptions that allow for a comprehensive understanding of the results. The figures are effective in showing the data, enhancing the overall clarity of the manuscript.
Upon review, no major technical or methodological issues were found. The authors have successfully demonstrated the relevance of their work within an in vitro context, contributing valuable knowledge to the field. However, it is important to note that the study appears to have limited direct translatability to human health or clinical applications at this stage. While the insights gained are significant for understanding cellular mechanisms in vitro, further investigation would be needed to establish their relevance in a clinical setting or for human therapeutic development.
Conclusion
We believe that the study meets the necessary technical standards for publication and provides meaningful contributions to in vitro research. From a technical perspective, the methodologies employed are sound, and the data support the authors conclusions. Therefore, we endorse this work for publication, particularly for its contribution to basic scientific knowledge and its potential to drive future studies.
Author Response
Summary
In this study, the combined effects of CpG ODN2006 and α-IgM antibodies, typically used for B cell activation, were examined on peripheral blood mononuclear cells (PBMCs). While previous research has focused on optimizing B cell activation using these agents, this work extends the analysis to include their impact on other immune cell types. Using flow cytometry and cytokine assays, the study identified that α-IgM, though effective in stimulating monocytes as evidenced by increased CD86 and HLA-DR expression, led to elevated monocyte-derived cytokine production. This activation, independent of B cells, suggests a potential drawback for clinical applications due to monocyte-driven side effects, even though it does not impede B cell activation. The authors conclude that while α-IgM can potentiate CpG ODN2006 in B cells, it may also cause undesirable monocyte activation, necessitating caution when considering this combination for therapeutic use.
Overall comment
The experimental design in this study is well-conceived. The statistical analyses applied are appropriate for the data. The data presentation is clear and well-organized, with detailed descriptions that allow for a comprehensive understanding of the results. The figures are effective in showing the data, enhancing the overall clarity of the manuscript.
Upon review, no major technical or methodological issues were found. The authors have successfully demonstrated the relevance of their work within an in vitro context, contributing valuable knowledge to the field. However, it is important to note that the study appears to have limited direct translatability to human health or clinical applications at this stage. While the insights gained are significant for understanding cellular mechanisms in vitro, further investigation would be needed to establish their relevance in a clinical setting or for human therapeutic development.
Conclusion
We believe that the study meets the necessary technical standards for publication and provides meaningful contributions to in vitro research. From a technical perspective, the methodologies employed are sound, and the data support the authors conclusions. Therefore, we endorse this work for publication, particularly for its contribution to basic scientific knowledge and its potential to drive future studies.
Reply: thank you very much for your detailed analysis of the manuscript and for the positive feedback!
Reviewer 5 Report
Comments and Suggestions for Authors
The study presents an original and interesting investigation evaluating the effects of combining CpG ODN2006 with a-IgM antibodies on B cell activation. Results indicate that PBMCs stimulation increased monocyte cytokine secretion.
To improve the manuscript quality,
Ensure that the keywords are sorted alphabetically.
Include SD values for the mean age of participants.
Ensure consistency in units and temperature notation (e.g., −180 ºC vs 37°C, or 200,000 cells vs 1500 rpm).
The authors should include whether the normal distribution and homoscedasticity were analyzed before the parametric test in the statistical section.
Author Response
he study presents an original and interesting investigation evaluating the effects of combining CpG ODN2006 with a-IgM antibodies on B cell activation. Results indicate that PBMCs stimulation increased monocyte cytokine secretion.
To improve the manuscript quality,
Ensure that the keywords are sorted alphabetically.
Include SD values for the mean age of participants.
Ensure consistency in units and temperature notation (e.g., −180 ºC vs 37°C, or 200,000 cells vs 1500 rpm).
The authors should include whether the normal distribution and homoscedasticity were analyzed before the parametric test in the statistical section.
Reply: Thank you for your positive feedback and suggestions. We included all suggested changes. The statistical analysis is indicated in the figure legends. If normal distribution was analysed, this is indicated in the text, otherwise not.